# Correlation between Biomarkers of Pain in Saliva and PAINAD Scale in Elderly People with Cognitive Impairment and Inability to Communicate

**DOI:** 10.3390/jcm10071424

**Published:** 2021-04-01

**Authors:** Vanesa Cantón-Habas, Manuel Rich-Ruiz, María Teresa Moreno-Casbas, María Jesús Ramírez-Expósito, Jose Manuel Martínez-Martos, María Del Pilar Carrera-González

**Affiliations:** 1Instituto Maimónides de Investigación Biomédica de Córdoba (IMIBIC), Universidad de Córdoba (UCO), Hospital Universitario Reina Sofía (HURS), Andalucía, 14004 Córdoba, Spain; n92cahav@uco.es (V.C.-H.); pcarrera@uco.es (M.D.P.C.-G.); 2Ciber Fragility and Healthy Aging (CIBERFES), 28029 Madrid, Spain; mmoreno@isciii.es; 3Nursing and Healthcare Research Unit (Investén-isciii), Instituto de Salud Carlos III, 28029 Madrid, Spain; 4Experimental and Clinical Physiopathology Research Group CTS-1039, Department of Health Sciences, Faculty of Health Sciences, University of Jaén, Campus Universitario Las Lagunillas, 23009 Jaén, Spain; mramirez@ujaen.es (M.J.R.-E.); jmmartos@ujaen.es (J.M.M.-M.)

**Keywords:** dementia, pain, pain assessment in advanced dementia (PAINAD), biomarkers, nursing

## Abstract

The pain assessment in advanced dementia (PAINAD) appears to be a clinically useful tool. However, the salivary determination of tumor necrosis factor receptor type II (sTNF-RII) and secretory IgA (sIgA) as pain biomarkers is still incipient. The aim was to correlate the PAINAD score with sTNF-RII and sIgA biomarker levels in the saliva of patients with advanced dementia. In this regard, a cross-sectional study was conducted. The sample consisted of 75 elderly patients with a clinical diagnosis of dementia and a global deterioration scale (GDS) score of 5 to 7. The PAINAD scale was determined by a previously trained professional and the collection of salivary samples was performed using the passive secretion method. Human sTNF-RII and sIgA using ELISA kits. The results showed a correlation between the PAINAD scale (numeric, binary, and recoded) and sTNF-RII and sIgA (*p* < 0.001). No association between the sociodemographic and clinical variables and the PAINAD scale was found (*p* > 0.05). Between 97.3% and 96.2% of patients with pain on the PAINAD scale also showed pain based on the sTNF-RII levels; in all of them, sIgA levels did not fit the logistic models. Therefore, the correlation highlights the usefulness of this scale and confirms the usefulness of sTNF-RII and sIgA as biomarkers of pain.

## 1. Introduction

Pain has been defined as “an unpleasant sensory and emotional experience associated with, or resembling that associated with, actual or potential tissue damage” [1]. Therefore, pain is not just a sensory process (nociception) [2]. It also involves cognitive and emotional factors influencing the expression of this experience [3,4].

Pain is common in older people because of the many chronic diseases that often affect this population group, such as osteoporosis, depression, cancer, or cardiovascular diseases [5]. It is estimated that approximately 50% of community-dwelling elders [6,7] and 80% of institutionalized elders have pain [8]. However, despite its high prevalence among the older population, it is a frequently underdiagnosed and undertreated [9,10]. This undertreatment is more marked among older people with dementia because of the difficulty in detecting it, and even more when the person cannot communicate effectively verbally due to disease progression [11,12]. In this case, primary caregivers and health care professionals are responsible for discerning whether or not the person is suffering pain [13].

Therefore, tools such as self-reports can be used to assess the presence and intensity of pain in early and even moderate stages of dementia, when the person still can abstract reasoning. In contrast, these pain assessment tools may yield biased results in patients with advanced dementia due to patients’ lack of ability to understand the concepts assessed through these tools and, in some cases, even to communicate this pain [14,15]. Consequently, the use of self-reports is restricted to initial or moderate stages of dementia, in the latter, provided that verbal communication skills are intact [16]. Thus, pain assessment tools based on direct patient observation emerged to mitigate pain under diagnostics in patients with dementia and the inability to communicate [17,18]. Today more than 24 tools are available that are elaborated and validated for this purpose. Of these tools, the PAINAD scale is considered the most suitable tool for the clinical setting due to its simplicity of use, extension, and psychometric properties [12,19]. However, this promising scale has moderate internal consistency and validity. It requires that the assessor has received specific training to properly assess the pain of the person with advanced dementia [20]. These limitations reinforce the need to supplement pain assessment in people with dementia and communication disabilities.

The determination of pain biomarkers is an innovative and useful way of assessing pain, especially in saliva, as a non-invasive and cost-effective test. Specifically, some biomarkers of pain, such as salivary cortisol, salivary amylase, testosterone, secretory IgA (sIgA), or tumor necrosis factor receptor type II (sTNF-RII), have been determined in previous studies [21,22,23,24,25]. However, among these, sIgA and sTNF-RII have been shown to be the most reproducible in healthy people [26], also being used to determine pain in surgical patients [27].

In this context, the potential correlation between the levels of biomarkers (sIgA and sTNF-RII) and the levels of pain reported by the PAINAD scale could be used in two directions: On the one hand, to reinforce the usefulness of this scale as a valid tool in the healthcare setting; and on the other hand, to suggest the determination of these biomarkers as a complementary method for the evaluation of pain in these patients.

## 2. Objectives

Correlate the scores of the PAINAD scale with the levels of pain biomarkers sTNF-RII and sIgA in saliva samples from patients with cognitive impairment.

## 3. Materials and Methods

### 3.1. Study Design

This has been a cross-sectional study performed between May 2018 and June 2020.

### 3.2. Study Setting

The Andalusian network of Primary Healthcare centers (two health districts from two provinces of Andalusia), four nursing homes and a center specifically dedicated to the care of dementia patients participated in the study.

### 3.3. Participants and Selection Criteria

The sample size was calculated for a correlation magnitude of r = 0.3, a statistical confidence interval (CI) of 95%, a statistical power of 80%, a unilateral approach, and 10% losses. The result sample size was 75 subjects.

The following inclusion criteria was required for participants:Being 65 or older.Having a medical diagnosis of dementia or Alzheimer’s disease-like dementia (AD) with a global deterioration scale (GDS) score between 5 and 7. Those patients who met DSM-V clinical criteria were diagnosed with dementia, and those who met the NINCDS/ADRDA (National Institute of Neurological and Communicative Disorders and Stroke/Alzheimer’s Disease and Related Disorders Association) criteria were classified as probable or possible AD patients.Being unable to communicate verbally.Having received health care at the community level for at least three months because of the dementia process. In the center dedicated to the care of patients suffering from AD, the included patients were those who have used this service for at least three months.Having an informed consent signed by a relative or legal representative for the patient’s inclusion in the study.

All subjects participating in the study were recruited consecutively by the interventional nursing, selecting those who met the previously mentioned criteria from among the subjects they care for in their health care institutions.

### 3.4. Study Measures and Data Collection

The main variables were the scores of the Spanish version of PAINAD and the values in saliva of the sTNF-RII and sIgA biomarkers. To assess the level of pain through the PAINAD scale, a researcher completed a book for data collection, having previously received training to ensure the reliability and reproducibility of the data. Likewise, the researchers received training with regard to how to perform the collection of the saliva sampler by using the passive secretion method [28]: Subjects were not allowed to practice physical exercise or to eat, ingest any drinks (except for water), chew gum, brush their teeth, or consume caffeine during the previous hour preceding the sample collection process.Five minutes before the collection of the sample, as a way to reduce the contamination of saliva with food debris, the subjects were asked to rinse their mouth with clean water.Right before starting the sample collection, any saliva present in the mouth at that time should be swallowed.Afterwards, the saliva accumulated in their mouths for 5 min was deposited in a collection tube, 1 mL being the minimum volume required. In the case of the 5 mL collection tube being filled before those 5 min, the corresponding amount of time elapsed was recorded.

Saliva samples were collected under supervision in a clinical setting, between 09:00 am and 10:00 am and ensuring it always was done before the intake of the morning medication. After collection, the samples were put under refrigeration and subsequently frozen (at −80 °C, to be concise) and kept in that state until the analysis was carried out.

In addition, sociodemographic variables (such as gender, age, and marital status) and clinical variables related to pain were collected from the patient’s medical records. Furthermore, the GDS and the level of autonomy in basic activities of day-to-day life, measured by the Barthel index, were determined by the research team at data collection or, in the cases in which that information was available (and not older than three months), it was collected from the clinical record.

### 3.5. Determination of sTNF-RII and IgA

ELISA kits were used to determine sTNF-RII and sIgA levels in saliva. Specifically, sTNF-RII levels were measured through the Human sTNF-RII Quantikine ELISA kit (R & D Systems, Minneapolis, MN, USA) and sIgA values were determined utilizing the sIgA ELISA kit (Salimetrics LLC, State College, Pennsylvania, PA, USA). In addition, Bradford’s method was the one used for the determination of total protein levels in saliva, in order to standardize the data. 

### 3.6. Data Analysis

Descriptive statistics were used to characterize the sample. In this sense, for quantitative variables, the following metrics were calculated: mean and its corresponding 95% CI, standard deviation (SD), minimum, maximum, interquartile range (IQR), median, and the number of observations. On the other hand, regarding qualitative variables, the corresponding frequency distributions were determined (both absolute and relative frequencies).

The matrix of polyserial correlations and the appropriate hypothesis contrasts were the tools of choice to evaluate the association between the levels of the biomarkers under study and the PAINAD scale.

Then, polyserial correlations and the χ^2^ coefficient with their corresponding significance levels were obtained to evaluate the potential relationships of demographic and clinical variables with biomarkers and PAINAD scores. 

Finally, in the case of detecting some kind of relationship between these variables, they were included in all logit and logistic multinomial models, in order to evaluate their relative influence on the fact that pain appears or not, and its corresponding degree.

This logit model was utilized with the goal of stablishing which saliva biomarker affects more the pain condition, and multinomial logistic models were considered to evaluate which of the biomarkers had the biggest influence on pain levels. 

The statistical package SPSS (V.22, IBM, Armonk, NY, United States) was employed for the majority of calculations mentioned previously, and the R software to determine the coefficients of polyserial correlations. Through all this statistical analysis, a significance of 5% was assumed.

### 3.7. Ethical Aspects

The study was carried out following the principles stablished in the Belmont report and the Helsinki Declaration (updated at the Seoul Assembly in 2008) for biomedical research. A Patient Information Sheet (PIS) was handed over to all family members or legal representatives of the candidates to offer them information regarding the general aspects of the study. Written informed consents were obtained and voluntarily signed by the patient’s relative or legal representative. All participants were allowed to withdraw consent to participate at any time during the development of the process.

Data confidentiality was always guaranteed, and so was the subject’s anonymity. In this sense, the study received the permission of all participating centers, as well as the approval of the Ethics Committee for Research of Andalusia (Acta nº 271, ref. 3672, approved on 5 December 2017).

## 4. Results

### 4.1. Sample Characteristics

A total of 75 participants were studied, of whom 59 (78.7%) were women. The overall mean age was 84.41 (7.44) years, 95% CI (82.70–86.13), and the range was 65–95 years. Regarding the marital status, a total of 68% of the sample were surviving spouses. Concerning the place of residence, 64 subjects lived in a rural area. In addition, 69.3% were institutionalized. 

Regarding the diagnosis of dementia, 69.33% had AD, which is, therefore, the most frequent type of dementia. Similarly, regarding the level of cognitive impairment, 34 subjects had a GDS 7. Furthermore, as for the level of independence for performing basic activities of daily living, the median Barthel index score was 10 (max = 80 and min = 0). 

Only 34 subjects of the total sample were prescribed analgesic treatment for their pain. The median of the PAINAD scale was 0 (max = 8 and min = 0), 95% CI (0.96–1.95). Furthermore, the median sTNF-RII (pg/mg protein) was 2.56 (max = 684.28 and min = 0.0162) 95% CI (5.84–49.28), whereas the median sIgA (ng/mg protein) determination in 54 of the participants was 5285.60 (max = 157440.83 and min = 6.49), 95% CI (4439.82–18905.34). Table 1 shows in detail the sociodemographic and clinical characteristics of the study subjects. In addition, Figure 1 and Figure 2 show the PAINAD numerical scores and the levels of sTNF-RII and sIgA, respectively.

### 4.2. Correlation between PAINAD Scale and Sample Characteristics

Concerning the possible association between the PAINAD variable (both the numerical version of the PAINAD variable, and the binary version and its recoding as: no pain, possible pain, and pain) and the rest of the sociodemographic variables of the categorical type, no correlation was found in any case (*p* > 0.05).

The degree of relationship between PAINAD (numeric) and the rest of the sociodemographic and clinical numerical variables, measured through the polyserial correlation coefficient, demonstrated a lack of relationship between the variables considered. Specifically, the results showed no relationship between PAINAD and age (*p* = 0.20), time consuming analgesics (*p* = 0.94), and Barthel index score (*p* = 0.21).

Similarly, no relationship has been found between PAINAD (in its binary recoding) and age (*p* = 0.11), time consuming analgesics (*p* = 0.44), and level of dependence for basic activities of daily living according to the score of the Barthel index (*p* = 0.21), based on the calculation of the polyserial correlation coefficient.

Similarly, the polyserial correlation coefficient showed no association between recoded PAINAD (no pain, possible pain, and pain) and age (*p* = 0.13), time consuming analgesics (*p* = 0.50), and Barthel index score (*p* = 0.37).

Therefore, the presence or absence of pain (in all grades) is independent of sex, age, marital status, area (rural/urban), institutionalization, type of dementia, time consuming analgesics, GDS, and degree of dependence according to the Barthel index. 

### 4.3. Correlation between PAINAD Scale and Saliva Biomarkers

The polyserial correlation coefficient, both following a simple two-step method and based on the maximum likelihood estimates method, was used to measure the degree of correlation between the PAINAD variables (numeric, binary, and recoded) and the levels of sTNF-RII and sIgA. In the two-step method, the value of the association between PAINAD (numeric) and sTNF-RII and sIgA was 0.76 (*p* < 0.001) and 0.56 (*p* < 0.001), respectively; whereas the value based on maximum likelihood estimation was 0.99 (*p* < 0.001) for sTNF-RII and 0.58 (*p* < 0.001) for sIgA.

In addition, in the two-step method, the associations between the binary-coded PAINAD scale score and the sTNF-RII and sIgA levels were 0.85 (*p* < 0.001) and 0.72 (*p* < 0.001), respectively. Additionally, according to the likelihood estimation, this ratio was 0.99 (*p* < 0.001) for sTNF-RII and 0.94 (*p* < 0.001) for sIgA.

Moreover, in the simple two-step method, the relationship between the recoded PAINAD variable and sTNF-RII was 0.79 (*p* < 0.001), whereas it was 0.99 (*p* < 0.0001) in the maximum likelihood estimation. In turn, the correlation between recoded PAINAD and sIgA was 0.70 (*p* < 0.001) in the two-step model and 0.95 (*p* < 0.001) according with the maximum likelihood estimate.

Therefore, all correlations can be considered high and statistically significant, although in all cases, the largest correlation was between the PAINAD score (numeric, binary, and recoded) and the sTNF-RII biomarker.

### 4.4. Regression Models

A binary logistic model was applied using a conditional step-by-step forward procedure to assess which of the two biomarkers had the largest influence on determining the presence of pain (no pain) in study subjects. In this model, the dependent variable was the binary PAINAD variable (0 no pain, 1 pain), and the sTNFR-II and sIgA levels were considered independent variables. The procedure excluded sIgA as a predictor variable in the model. Thus sTNF-RII was determined as the most influential variable for the prognosis of pain. Data obtained in the analysis are shown in Table 2.

The PAINAD variable was then recoded into three categories (no pain, possible pain, and pain). In this case, a multivariate logistic model was applied to again determine the most influential indicator. Table 3 and Table 4 show that again the variable sIgA was ruled out from the model (sig. > 0 0.05).

In relation to the original PAINAD variable, Table 5 and Table 6 show that the results in determining the multivariate logistic model were similar to the above mentioned, ruling out the sIgA variable again.

## 5. Discussion

Our results support the usefulness of the PAINAD scale for assessing pain in patients with advanced dementia, given the correlation between the sTNF-RII and sIgA measurements and the score on the PAINAD scale. Therefore, the PAINAD scale was confirmed as a simple and feasible tool, but with remarkable precision, that allows to assess pain in people with dementia and inability to communicate and thus to mitigate the characteristic underdiagnosis and undertreatment in these patients [29,30]. However, it is necessary to persuade the health care professional who will conduct the assessment mentioned above to acquire training for its correct use [20,31].

Similarly, the determination of pain biomarkers in fluids such as the saliva is a promising strategy [32], especially for patients with advanced dementia and the inability to communicate, where direct reporting of perceived pain is impossible. In addition, saliva is a biological sample whose accessibility is significantly less invasive than blood determinations. In this sense, the determination of both sTNF-RII and sIgA in saliva, in line with the results of other authors, meets the requirements to be considered appropriate for its implementation in clinical practice because it is safe, relatively simple, reproducible, and non- invasive [26]. However, there is a need to clarify the sometimes existence of difficulty in taking salivary samples, such as xerostomia associated with advanced dementia [33,34].

Regarding the correlation between both biomarkers (sTNF-RII and sIgA) and the PAINAD scale, based on the results achieved, both should be considered as biomarkers of pain. Specifically, regarding sTNF-RII levels, it has been shown that between 97.3% and 96.2% of patients with pain on the scale would also have pain according to sTNF-RII. This result would probably be related to the pain-modulating function of some cytokines [35], including tumor necrosis factor-alpha (TNF-α), which would explain why an increase in sTNF-RII, which is a TNF-α receptor, involves an increase in pain based on the PAINAD scale. However, it should be noted that, in addition to this pain-modulating function, TNF-α is a pro-inflammatory cytokine involved in the immune response [36]. Additionally, neuroinflammation plays a significant role in developing dementia and AD [37,38,39,40], which implies high TNF-α levels at baseline in both pain and no pain patients.

The results, therefore, confirm the usefulness of the PAINAD scale for pain assessment in patients with advanced dementia. Moreover, our results point to the determination of sTNF-RII in saliva as an effective method by itself and complementary to the PAINAD scale, if it is desired to objectify the assessment.

Furthermore, the correlation found in our study between the PAINAD scale values and the sIgA levels has also previously been reported by Sobas et al. (2016) in healthy patients, which would allow us to establish a link between pain and the pain biomarker sIgA [26]. However, these studies do not point to the mechanisms associating sIgA with pain, and our study does not show it either. The role of sIgA as an immunoglobulin involved in protecting the oral mucosa and, consequently, its relationship with oral infections and inflammatory processes is known [41]. Recent studies also report an association between periodontal diseases and dementia processes [42,43,44]. However, we cannot demonstrate that this relationship occurs in our study because no periodontal diseases were identified in the subjects studied.

The absence of a relationship between pain and sociodemographic characteristics (sex, age, marital status, area and institutionalization) could be conditioned by the characteristics of our sample (patients with advanced dementia). The associations found in previous studies seem to be related, at least in part, to the psychological component of pain (the thoughts the person has about what happens to him/her and what he/she feels) [45,46].

The lack of association with clinical characteristics may be because our study subjects represent a very homogeneous population (with significant cognitive impairment and a significant level of dependence for basic activities of daily living). This would have made it difficult to recognize such an association.

Finally, the lack of association between analgesic use and the PAINAD scale could be related to the scarce and limited analgesic treatment of the study subjects due to the frequent underdiagnosis and undertreatment of pain in this group that, often, is constricted to the first step of analgesia in the World Health Organization (WHO) classification [47,48]. Therefore, new studies are required to truly assess whether undertreatment of pain improves after the inclusion of these tools (PAINAD scale and saliva biomarkers) for pain assessment in clinical practice.

### Limitations

The xerostomia present in some subjects has hindered the collection of enough salivary sample volume for measuring both biomarkers in some subjects. Therefore, the sample size was relatively lower (*n* = 54) in the measurement of sIgA. However, statistical analysis has shown that the number of subjects was sufficient to achieve significant results.

## 6. Conclusions

The correlation between the PAINAD scale and the sTNF-RII and sIgA pain biomarkers reinforce the usefulness of this scale in assessing pain in people with dementia and inability to communicate. Similarly, the sTNF-RII and sIgA salivary biomarkers are suggested as efficient and complementary tools of utmost applicability for pain assessment, and sTNF-RII is most accurately predicting pain in people with advanced dementia, according to the PAINAD scores.

## Figures and Tables

**Figure 1 jcm-10-01424-f001:**
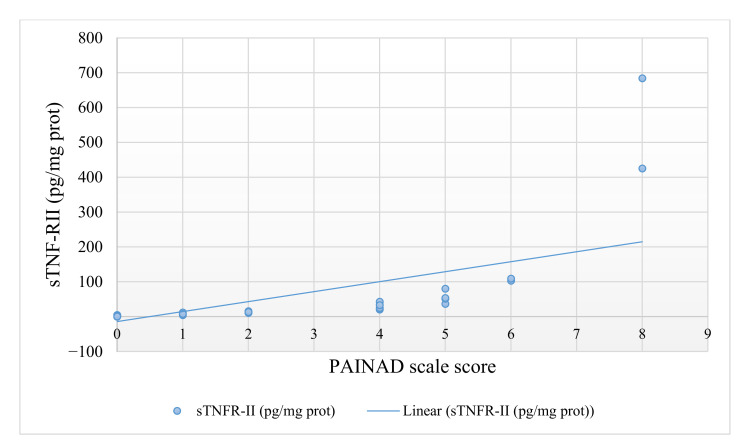
Association between PAINAD numerical score and sTNF-RII (pg/mg protein) values.

**Figure 2 jcm-10-01424-f002:**
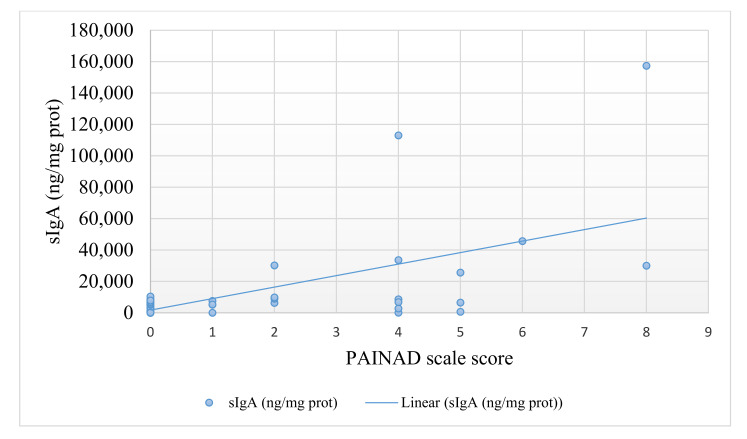
Association between pain assessment in advanced dementia (PAINAD) numerical score and sIgA (ng/mg protein) values.

**Table 1 jcm-10-01424-t001:** Description of the sample.

Variable	Total *n* = 75
Sociodemographic variables	
Age x (SD)	84.41 (7.44)
Sex *n* (%)	
Woman	59 (78.7%)
Man	16 (21.3%)
Marital status *n* (%)	
Single	6 (8%)
Married	16 (21.3%)
Widowed	51 (68%)
Divorced	2 (2.7%)
Institutionalized *n* (%)	
Yes	52 (69.3%)
No	23 (30.7%)
Area of residence of the sample *n* (%)	
Rural	11 (14.7%)
Urban	64 (85.3%)
Clinical variables	
Type of dementia *n* (%)	
Alzheimer’s disease	52 (69.33%)
Vascular dementia	8 (10.67%)
Primary degenerative dementia	2 (2.67%)
Mixed dementia	13 (17.33%)
Level of cognitive impairment ^b^ *n* (%)	
GDS 5	13 (17.33%)
GDS 6	28 (37.33%)
GDS 7	34 (45.33%)
Level of dependency for basic activities of daily living ^c^ *n* (%)	
Moderate dependence	7 (9.3%)
Severe dependence	18(24%)
Total dependence	50 (66.7%)
Prescription of analgesics *n* (%)	
Yes	34 (45.33%)
No	41 (54.67%)
Time consuming analgesics (days) M_e_ (maximum-minimum) ^a^	471 (4321–5)
PAINAD x (SD)	1.45 (2.15)
sTNF-RII (pg/mg protein) M_e_ (maximum-minimum) ^a^	2.56 (684.28–0.0162)
sIgA (ng/mg protein) (*n* = 54) M_e_ (maximum-minimum) ^a^	5285.60 (157440.83–6.49)

^a^ The median has been calculated given the robustness of this analysis and because the range of possible values is very wide.^b^ Assessed using the global deterioration scale (GDS).^c^ Assessed using the Barthel index.

**Table 2 jcm-10-01424-t002:** Binary logistic model between the binary PAINAD variable and the sTNFR-II and sIgA levels.

Variables in the Equation						
Dependent Variable: Binary_PAINAD	B	Standard Error	Wald	df	Sig.	Exp(B)
sTNF-RII (pg/mg prot) Constant	2.177	1.301	2.801	1	0.094	8.817
	−10.469	6.118	2.928	1	0.087	0.000

Nagelkerke R^2^ 0.963. Cox & Snell R^2^ 0.716 Cut-off point 0.5% of well-classified cases 97.3 model sig. 0.000.

**Table 3 jcm-10-01424-t003:** Multivariate logistic model between PAINAD score (three categories) and sTNF-RII and sIgA.

Parameter Estimates									
Recoded PAINAD ^a^		B	Standard Error	Wald	df	Sig.	Exp(B)	95% Confidence Interval for Exp(B)	
								Lower Limit	Upper Limit
Possible pain	Interception	−68.332	2827.925	0.001	1	0.981			
	sTNF-RII	10.264	0.702	213.580	1	0.000	28667.235	7237.641	113546.716
	sIgA	0.001	1.530	0.000	1	0.999	1.001	0.050	20.072
With pain	Interception	−77.191	2827.924	0.001	1	0.978			
	sTNF-RII	10.915	0.000		1	0.000	54977.698	54977.698	54977.698
	sIgA	0.002	1.530	0.000	1	0.999	1.002	0.050	20.077

^a^. The reference category is: no pain.

**Table 4 jcm-10-01424-t004:** Parameters for adjusting the multivariate logistic model between the PAINAD scale (three categories) and sTNF-RII and sIgA.

Likelihood Ratio Comparison				
Effect	Model Fit Criteria	Likelihood Ratio Comparison		
	-2Log Likelihood of the Reduced Model	Chi-Square	df	Sig.
Interception	79.552	75.041	2	0.000
sTNF-RII (pg/mg prot)	66.945	62.433	2	0.000
sIgA (ng/mg prot)	5.343	0.831	2	0.660

Nagelkerke R^2^ 0.979. Cox & Snell R^2^ 0.790% of well-classified cases 96.2.

**Table 5 jcm-10-01424-t005:** Multivariate logistic model between the original PAINAD score and sTNF-RII and sIgA.

Parameter Estimates									
PAINAD ^a^ (0–10 Points) ^c^		B	Standard Error	Wald	df	Sig.	Exp(B)	95% Confidence Interval for Exp(B)	
								Lower Limit	Upper Limit
1	Interception	−61.291	3968.797	0.000	1	0.988			
	sTNF-RII	9.203	595.872	0.000	1	0.988	9923.542	0.000	^b^
	sIgA	0.001	0.738	0.000	1	0.999	1.001	0.236	04.255
2	Interception	−69.780	3968.804	0.000	1	0.986			
	sTNF-RII	9.801	595.872	0.000	1	0.987	18055.531	0.000	^b^
	sIgA	0.002	0.738	0.000	1	0.998	1.002	0.236	4.256
4	Interception	−164.123	8885.031	0.000	1	0.985			
	sTNF-RII	15.207	737.434	0.000	1	0.984	4020202.510	0.000	^b^
	sIgA	0.001	0.752	0.000	1	0.999	1.001	0.229	4.368
5	Interception	−333.143	20651.032	0.000	1	0.987			
	sTNF-RII	18.817	788.261	0.001	1	0.981	148569218.102	0.000	^b^
	sIgA	0.001	1.035	0.000	1	0.999	1.001	0.132	7.617
6	Interception	−393.108	20877.084	0.000	1	0.985			
	sTNF-RII	19.192	847.243	0.001	1	0.982	216275110.473	0.000	^b^
	sIgA	0.002	0.805	0.000	1	0.998	1.002	0.207	4.852
8	Interception	−414.142	0.000		1				
	sTNF-RII	19.308	852.322	0.001	1	0.982	242920504.208	0.000	^b^
	sIgA	0.001	0.964	0.000	1	0.999	1.001	0.151	6.622

^a^. The reference category is: 0. ^b^. A floating-point overflow occurred while calculating this statistic. Therefore, its value is set as a system loss. ^c^. Numerical score on the PAINAD scale, which can range from 0 (no pain) to 10 (maximum pain) points. Values 3, 7, 9, and 10 are absent as no subject obtained this score.

**Table 6 jcm-10-01424-t006:** Parameters for adjusting the multivariate logistic model between the original PAINAD and sTNF-RII and sIgA.

Likelihood Ratio Comparison				
Effect	Model Fit Criteria	Likelihood Ratio Comparison		
	-2Log Likelihood of the Reduced Model	Chi-Square	df	Sig.
Interception	178.552	174.056	6	0.000
sTNF-RII	110.324	105.827	6	0.000
sIgA	5.343	0.846	6	0.991

Nagelkerke R^2^ 0.993 Cox & Snell R^2^ 0.916% of well-classified cases 96.2.

## Data Availability

Data are available upon reasonable request.

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
