# Peer review of "Correlation between Biomarkers of Pain in Saliva and PAINAD Scale in Elderly People with Cognitive Impairment and Inability to Communicate"

_jcm, 2021, doi:10.3390/jcm10071424_

Round 1

Reviewer 1 Report

Please, see few comments in the attached pdf

Author Response

Firstly, we would like to thank the reviewers of the article for their comments, since their proposals for improving the article have unquestionably made it possible to achieve a better manuscript. After making the appropriate modifications we would like to comment in detail the changes we made, for each reviewer.

In response to reviewer 1's comments we would like to clarify the following:

  • We have taken into consideration the recommendations made regarding the language errors in the manuscript. In this respect, as you can observe in the document, the proposed changes have been addressed. Please excuse any inconvenience caused by these mistakes.
  • Regarding the missing abbreviations, they have now been defined in the abstract. As an additional note, about the GDS abbreviation required in the “study measures and data collection”, we made an exception and it has not been substituted by the words it stands for because the abbreviation had already been defined previously (in the previous section, entitled “participants and selection criteria”).
  • In the section called “participants and selection criteria”, we were asked to include the justification for the statistical analysis used. This calculation was carried out to estimate the significance of a correlation coefficient between two variables measured on the same subject. This estimation makes it possible to establish the correlation coefficient and its significance for a value other than 0.

Reviewer 2 Report

The study appears to be well planned and written in an understandable way. The bibliography is quite recent, more than half of the articles mentioned have been published in the last 5 years and is mostly related to the subject matter, but in several tables statistical analys is not clear (not easy to follow).

Comments

My big question is: are two (or one) biomarkers enough to define the presence of pain?

  • Line 33: thesentence  "in all of them, sIgA levels does not fit the models" is not very clear. Aside from the minor grammatical error concerning the plural of level and the conjugation of "to do",  it seems like the incongruity between sIgA levels and the established models to assess the pain is undesired, when in fact it's one of the objective of this study.
  • Line 174:a chart showing Bathel's score would make the low median value of 10 clearer.
  • Line 176:the same goes for PAINAD'S scale, which is mentioned multiple times throughout the article, but never shown. Since the possible audience of this paper might include professionals without any expertise in pain assessment, such as laboratorists, a chart would be usefull.
  • line177:there are no pictures showing Barthel and PAINAD's score among patients, nor their individual sTNF-RII and sIgA levels.
  • Line 178:why were the sIgA levels mesured only in 54 of the total 75 patients?
  • Line 213:check 72 (p>0.001)
  • Line 233:in Table 2 sIgA levels are mentioned, but not shown
  • Line 248: table 5 does not explain the meaning of 1-8
  • Line 265:instead of "less bloody than blood determinations" what about "less traumatic than blood determinations''?

  • In Results and Discussion, it is not clear ifsIgA is or not a  biomarker of pain.

  • Discussion. Do you think that administering painkillers would beget a sensible and sufficient reduction, meaning the therapy worked? If so, wouldn't you think further studies are needed to quantify this mitigation and the limits under which the pain is under control? Authors should be discuss this point.

Author Response

Please find attached a document with the response point to point to your suggestions
